# Nosocomial Infections in Adult Patients Supported by Extracorporeal Membrane Oxygenation in a Cardiac Intensive Care Unit

**DOI:** 10.3390/microorganisms11041079

**Published:** 2023-04-20

**Authors:** Simone Mornese Pinna, Iago Sousa Casasnovas, María Olmedo, Marina Machado, Miriam Juàrez Fernández, Carolina Devesa-Cordero, Alicia Galar, Ana Alvarez-Uria, Francisco Fernández-Avilés, Jorge García Carreño, Manuel Martínez-Sellés, Francesco Giuseppe De Rosa, Silvia Corcione, Emilio Bouza, Patricia Muñoz, Maricela Valerio

**Affiliations:** 1Servicio de Microbiología Clínica y Enfermedades Infecciosas, Hospital General Universitario Gregorio Marañón, 28009 Madrid, Spain; 2Servicio de Cardiología, Hospital General Universitario Gregorio Marañón, CIBERCV, 28007 Madrid, Spain; iagosousa1979@gmail.com (I.S.C.);; 3Instituto de Investigación Sanitaria Gregorio Marañón, 28009 Madrid, Spain; 4Facultad de Medicina, Universidad Complutense de Madrid, 28040 Madrid, Spain; 5Centro de Investigación Biomédica en Red de Enfermedades Respiratorias (CIBERES), Instituto de Salud Carlos III, 28029 Madrid, Spain; 6Department of Medical Sciences, Infectious Diseases, University of Turin, A.O.U. Città della Salute e della Scienza di Torino, 10124 Turin, Italysilvia.corcione@unito.it (S.C.)

**Keywords:** ECMO, venoarterial extracorporeal membrane oxygenation, nosocomial infections, cardiac intensive care unit, cardiovascular infections

## Abstract

The use of venoarterial (VA) extracorporeal membrane oxygenation therapy (ECMO) in patients admitted to cardiac intensive care units (CICU) has increased. Data regarding infections in this population are scarce. In this retrospective study, we analyzed the risk factors, outcome, and predictors of in-hospital mortality due to nosocomial infections in patients with ECMO admitted to a single coronary intensive care unit between July 2013 and March 2019 treated with VA-ECMO for >48 h. From 69 patients treated with VA-ECMO >48 h, (median age 58 years), 29 (42.0%) patients developed 34 episodes of infections with an infection rate of 0.92/1000 ECMO days. The most frequent were ventilator-associated pneumonia (57.6%), tracheobronchitis (9.1%), bloodstream infections (9.1%), skin and soft tissue infections (9.1%), and cytomegalovirus reactivation (9.1%). In-hospital mortality was 47.8%, but no association with nosocomial infections was found (*p* = 0.75). The number of days on ECMO (OR 1.14, 95% CI 1.01–1.30, *p* = 0.029) and noninfectious complications were higher in the infected patients (OR: 3.8 95% CI = 1.05–14.1). A higher baseline creatinine value (OR: 8.2 95% CI = 1.12–60.2) and higher blood lactate level at 4 h after ECMO initiation (OR: 2.0 95% CI = 1.23–3.29) were significant and independent risk factors for mortality. **Conclusions:** Nosocomial infections in medical patients treated with VA-ECMO are very frequent, mostly Gram-negative respiratory infections. Preventive measures could play an important role for these patients.

## 1. Introduction

Venoarterial (VA) extracorporeal membrane oxygenation (ECMO) is an advanced technique temporarily providing circulatory and ventilatory support to patients affected by severe cardiac and/or respiratory failure [1]. In recent decades, due to technological advances, the number of patients being supported by VA-ECMO has grown worldwide [2]. The implantation of VA-ECMO has been demonstrated to improve the outcome in patients with severe cardiac disease; nonetheless overall survival is still modest, and over half of patients are expected to die in hospital [3,4,5]. Patients supported by ECMO are exposed to a broad range of complications due to the underlying critical illness, the multiple invasive procedures and devices, including the ECMO cannulas, and the longer intensive care unit (ICU) stay [6,7]. Nosocomial infections (NIs) remain a major contributor to the excess of morbidity and mortality [8,9]. However, studies addressing the epidemiology of NIs on adult patients supported by VA-ECMO are scarce^10^. Notably, these studies were addressed in generic ICU wards. We sought to investigate the risk factors and the incidence and microbiology of NIs in patients who underwent VA-ECMO in a cardiac intensive care unit (CICU).

## 2. Materials and Methods

### 2.1. Setting and Patients

Ours was a retrospective study performed at Hospital General Universitario Gregorio Marañón, a 1200-bed tertiary hospital in Madrid, Spain. All adults (>18 years) receiving VA-ECMO >48 h for cardiac arrest or severe cardiogenic shock from January 2013 to December 2018 within the cardiovascular intensive care unit (CICU) were evaluated for inclusion in the study. The CICU has 12 beds, single-bed rooms, separated by walls, with a glass door that is kept closed. The air ventilation system is HEPA filtered. The ECMO is placed in a separate room where hemodynamic procedures are performed; in the case of complications, the patient is transferred to the cardiac surgery operating room. The nurse–patient ratio is 1:1 in CICU.

All the patients were followed by a multidisciplinary team with great expertise in infectious diseases in cardiac patients. The exclusion criteria were to have received ECMO support <48 h or VV-ECMO support. The patients included were followed up until death while on ECMO or up to 3 weeks after ECMO weaning.

### 2.2. Cannulation Characteristics

Central or peripheral cannulation was performed for ECMO implant. For central cannulation, cardiac surgery was required. The peripheral approach through the femoral artery and vein was the most commonly applied in the study and was performed by an interventional cardiologist. In both cases, the intervention was performed in the operating room. In patients who underwent ECMO implant, a short course (3 doses) of cefazolin were administered (or vancomycin in case of beta-lactam allergy); if an underlying infection was thought to be present at the time of cannulation in patients carrying risk factors for methicillin-resistant S. aureus (MRSA) infection, a beta-lactam/beta-lactamase inhibitor combination +/− vancomycin was started.

### 2.3. Antimicrobial Surveillance and Standard of Care

Peripheral lines and central catheters were coated with chlorhexidine-impregnated sponges at the insertion site. Nasal colonization for *S. aureus* was investigated through a nasal swabbing culture followed by molecular detection (Xpert^®^ SA Nasal Complete assay). In patients with *S. aureus* nasal carriage, decolonization with Mupirocin 2% nasal ointment for 5 days, and a daily chlorhexidine whole-body washing was performed before ECMO cannulation. If bronchial aspiration was suspected during orotracheal intubation (especially with prehospital cardiorespiratory arrest), a course of 5 days of amoxicillin-clavulanate was given to the patient or fluoroquinolone in the presence of a beta-lactam allergy. Since 2014, oral decontamination and selective digestive decontamination was used in patients under mechanical ventilation (MV) for more than 48 h and was maintained until MV finished. Oral decontamination was conducted with a paste formulated with: colistin 20 mg, nystatin 20 mg, tobramycin 30 mg, and vancomycin 40 mg, and it was administered once every 6 h. Selective digestive decontamination was conducted with a 10 mL solution formulated with nystatin 2.6 MUI and tobramycin 156 mg, administered by nasogastric tube once every 6 h. In patients colonized/infected with MRSA, ESBL, or carbapenemase-producing bacteria, contact isolation was performed. In patients infected with respiratory viruses, respiratory isolation was performed.

When an infection was suspected, an infectious disease specialist provided consult, before starting empirical antibiotic therapy.

### 2.4. Nosocomial Infections

Nosocomial infections (NIs) were defined according to the definitions proposed by the Centers for Disease Control and Prevention (CDC) [10]. NIs occurring >24 h after the initiation of VA-ECMO and within 48 h after VA-ECMO weaning were evaluated. Ventilator-associated pneumonia (VAP) in patients supported by ECMO was defined by at least one of the following: fever > 38.0 °C, leukocytosis > 12^9^/L, or leukopenia < 4 × 10^8^/L, and at least two of the following: new onset or worsening of pre-existing low respiratory purulent discharge, a new persistent infiltrate or worsening or a pre-existing one on chest radiography, with or without a positive Gram stain and culture of bronchoalveolar lavage or bronchial/tracheal aspirate) [11]. In contrast, the clinical and laboratory findings of VAP in the absence of radiological changes were defined as ventilator-associated tracheobronchitis [12]. Bloodstream infection (BSI) was defined as the isolation of bacterial or fungal pathogens from one or more blood cultures [13]. A single positive blood culture for coagulase-negative staphylococci (CoNS) or another common skin contaminant, in the absence of signs of systemic inflammatory response was defined a contamination. Cytomegalovirus (CMV) disease was defined as the detection of CMV-DNA ≥ 1000 UI/mL and clinical symptoms compatible with organ damage. A causative organism was defined as multidrug resistant, according to the CDC definition [14].

### 2.5. Variables Analyzed

The epidemiology, risk factors, and frequency of NIs were evaluated through the hospital records. We assessed the overall CICU and hospital lengths-of-stay (LOS) and the major surgery before VA-ECMO placement. We also recorded the overall mortality and attributable mortality to NIs. The infection rate was calculated as the number of infections divided by the total number of ECMO days during the time period *1000 in order to describe the number of events per 1000 ECMO days.

### 2.6. Statistical Analysis

The characteristics of patients supported by ECMO with and without NIs were compared. Statistical analyses were performed using SPSS 23.0 (IBM Corp. Released 2015. IBM SPSS Statistics, Version 23.0. IBM Corp. Armonk, NY, USA). Continuous variables were analyzed using a nonparametric tests (Mann–Whitney U or Kruskal Wallis); conversely, categorical variables were analyzed using Fisher’s exact test. All tests were two-tailed, and the result was considered significant at a value of *p* < 0.05. Continuous variables were reported as the median and interquartile range (IQR). A logistic regression model was applied for univariate and multivariate analyses to determine the independent predictive factors associated with NIs during ECMO. The variables included in the multivariate analysis were selected based on the results from the univariate analysis and presumptive association based on previous data. A mortality analysis was also performed using Kaplan–Meier; an “event” was defined as all-cause in-hospital mortality.

### 2.7. Ethics

This study was conducted in accordance with the ethical standards of the local institution’s committee (MICRO.HGUGM.2019-010), approved on 4 July 2019.

## 3. Results

During the six-year study period, 98 patients were treated with VA-ECMO. Of these, 29 patients received VA-ECMO for less than 48 h because of death or early VA-ECMO weaning, or they received VV-ECMO and were excluded. Sixty-nine patients underwent a total of 534 VA-ECMO days for cardiac support and constituted our study population. The general characteristics of all the patients with ECMO and a comparison of those with and without infection are shown in Table 1. The median age was 58 years (IQR 50.5–62.8), and most of them were males (82.7%). A majority of the patients (75.4%) had previous cardiac disease before CICU admission. The primary causes for VA-ECMO support were cardiogenic shock (55.1%) and cardiac arrest (30.4%). Of these, 13% of patients received VA-ECMO as a “bridge” to heart transplantation.

Overall, 30.4% of patients were previously colonized, all of them with nasal carriage of *S. aureus*. No prior colonization by ESBL or carbapenem-resistant Enterobacteriaceae was detected. A total of 10 (14.5%) of the patients had evidence of infection before ECMO placement.

Overall, 29 patients (42%) developed 34 episodes of NIs after ECMO placement. These figures correspond to 63.7 infectious episodes/1000 ECMO days and to an infection rate of 0.92. The median interval from ECMO initiation to the development of NIs was 1.8 days. Regarding the number of infections per patient, 25 (86.2%) developed one episode of NI, whilst 3 (10.3%) suffered two events, and 1 patient (3.4%) developed three episodes of NIs.

### 3.1. Risk Factors Associated with Infection Development

When comparing patients with and without infection (Table 1), univariate analysis revealed that there were no significant differences in terms of age, sex, underlying diseases, and pre-ECMO implantation variables. We only found that pre-existing cardiac disease was most frequently observed among the infected patients (93.1% vs. 63.5%, *p* = 0.004). The immunosuppressive status, Charlson comorbidity score, and SOFA score did not differ between groups (*p* = 0.73, *p* = 0.55, *p* = 0.53, respectively), nor did the number of days of mechanical ventilation before ECMO (*p* = 0.90) (Table 1). No differences were found between the infected and noninfected patients in terms of the type of indication for which ECMO was implanted. As for the variables related to the time of ECMO implantation, shown in Table 2, there were no differences between the infected and noninfected patients.

However, the patients who developed an infection also had more complications during ECMO than those who did not (53.7% vs. 46.3%, *p* = 0.044), mostly cerebrovascular events (13.8% vs. 0%, *p* = 0.031) (Table 2). The NIs are shown in Figure 1. Furthermore, the duration of ECMO tended to be longer in infected vs. noninfected patients (median days, 5.73 vs. 7.20, *p* = 0.053) (Table 2). After multivariate analysis, the variables significantly associated with NIs were the number of days on ECMO (OR: 1.14, 95% CI 1.01–1.30, *p* = 0.029) and the complications during ECMO (OR: 3.8 95% CI = 1.05–14.1).

### 3.2. Type of Nosocomial Infection and Microbiology

Lower respiratory-tract infections (LRTI) were the most common type of infection (22/34, 64.7%) (Table 3). We found 19/34 (55.9%) episodes of VAP and 3/34 (8.8%) episodes of tracheobronchitis (36.6 episodes of VAP and 5.6 episodes of tracheobronchitis per 1000 ECMO days). VAP occurred a mean of 6.7 days after mechanical ventilation. Among patients with low-tract respiratory infections, a significant microbiological identification was obtained in 20/22 samples. Among them, Gram-negative grew in 11/19 samples (57.9%).

A total of three patients developed a bloodstream infection (BSI) for an overall BSI rate of 5.6/1000 ECMO days. We found three skin and soft tissue infection (SSTI) episodes (5.6/1000 ECMO) and one episode each of intra-abdominal infection, *Clostridium difficile* infection, and urinary tract infection (rate 1.9/1000 ECMO days, each). In three patients (rate 5.6/1000 ECMO days), a significant CMV reactivation was observed.

Septic shock occurred in two patients as a result of an episode of VAP and BSI, respectively. A total of 5/34 (14.7%) episodes of infections in four patients were due to MDR isolates (two episodes of VAP due to MRSA and *Enterobacter aerogenes,* respectively, and two BSI and one SSTI due to coagulase negative *Staphylococcus* (CoNS). The microorganisms associated with the NIs are shown in Table 3.

From the three episodes of BSI, one was due to *P. aeruginosa* and two were due to CoNS. Finally, two episodes of SSTI were caused by CoNS and one by *Morganella morganii*.

Overall, Gram-negatives were responsible for 42% of NIs, including 10.5% non-fermentative GNB, confirming their pivotal role in VAP [15]. Gram-positive microorganisms, mainly from the Staphylococcus genus were responsible for 26% of NIs.

### 3.3. Outcomes

Overall, the median ICU stay was 14 days, and the median hospital stay was 20.3 days. In-hospital and ICU length of stay were not significantly different between the infected and noninfected patients (Table 2).

Overall, the in-hospital mortality rate was 47.8%%, and the mortality rate during ECMO was 39.1%. The Kaplan–Meier analysis did not show a difference in mortality between the infected and noninfected patients (50% vs. 46.2% *p* = 0.75, Figure 2). The attributable mortality due to VAP and BSI in our population was 13%. In order to analyze the risk factors associated with mortality we compared the survivors vs. non-survivors (Table 4). Regarding variables prior to ECMO implantation, in the univariate analysis, the non-survivors had a higher baseline serum Cr (median 1.24 mg/dl) compared with the survivors (median 0.93) (*p* = 0.006). As for the variables during ECMO therapy, the non-survivors had a higher transfusion requirement during ECMO (62.5% vs. 37.8%, *p* = 0.055) and a higher lactic acidosis 2 h after starting ECMO (6.3 vs. 5.0, *p* = 0.001). After ECMO placement, the non-survivors experienced significantly more strokes (53.1% vs. 8.3%, *p* < 0.047) and acute kidney failure (53.1% vs. 8.3%, *p* < 0.0001) than the survivors. Multivariate analysis revealed that a higher baseline creatinine value (reflecting renal failure prior ECMO placement) was strongly associated with death (OR: 8.2 95% CI = 1.12–60.2). Moreover, a higher blood lactate level at 4 h, reflecting lactic acidosis, (3.9 vs. 1.7, *p* = 0.001), after ECMO initiation was a significant and independent risk factor of mortality (OR: 2.0 95% CI = 1.23–3.29).

## 4. Discussion

We described the epidemiology and outcomes of nosocomial infections of 69 patients supported by VA-ECMO in a single CICU for cardiac arrest or cardiogenic shock. We found that 42% of patients developed at least one episode of nosocomial infection (Infection rate of 71.2/1000 ECMO days). The reported rate of ECMO-related NIs varies widely depending on age, nature, and case selection with a trend towards greater incidence in adults than in pediatric patients [15,16,17]. Both in pediatric and adult populations, the incidence of NIs increase proportionally to the rate of patients with underlying cardiac disease; however, data focusing on adults supported by VA-ECMO are scarce [18,19,20,21,22]. Our results are consistent with previous studies of adult patients who underwent VA-ECMO, which reported an incidence of NIs of 23% to 64% and an infection rate that varied from 43 to 75.5 episodes/1000 ECMO days [9,16,22,23,24]. We observed a trend toward an association among ECMO duration and the risk of developing Nis (7.20 vs. 5.73 days, *p* = 0.053), but it was not significant [9,15,22].

In line with previous analyses [16,21,23,25], we found that VAP was a frequent ECMO-related infection, occurring in 27.5% of patients (36.6 episodes/1000 ECMO days) after a median time of 6.7 days of MV. The largest retrospective study of patients who underwent VA-ECMO for refractory cardiogenic shock was published by Schmidt et al. [23], and they found that VAP was the most frequent infection and accounted for 55% of all NI episodes. Furthermore, Bouglè et al. [26] showed that the incidence of VAP was 56% among patients undergoing VA-ECMO with high rate of recurrence.

It is noteworthy that although the indications to ECMO, length of extracorporeal support, and surveillance measures were similar, in the studies of Bouglè and Schmidt a higher SOFA score at baseline and greater proportion of immunocompromised individuals suggested a more severe organ disfunction and higher risk of developing NIs [23,26]. On the other hand, Hsu et al. reported a surprising low rate of NIs, including 3.5% of respiratory infections in 114 patients supported by ECMO for cardiogenic shock (85% of them with VA-ECMO) [21]. Nevertheless, in that study, 74% of patients were on antibiotics for the entire length of ECMO support.

The epidemiology of NIs among VA-ECMO-supported patients did not differ from that described in other ICUs with a predominance of Gram-negatives. We found that Gram-negatives were responsible for 42% of NIs, including 10.5% non-fermentative GNB, confirming their pivotal role in VAP, regardless of ECMO support [27]. Surprisingly, we found a low rate of MDR organisms compared to other studies, manly among Gram-positives, and only one episode of VAP due to MDR *E. cloacae* was noted [28].

Gram-positive microorganisms, mainly from the Staphylococcus genus, were responsible for 26% of NIs. Three of our patients suffered a cytomegalovirus (CMV) reactivation, one of them was an immunosuppressed host (liver transplant recipient), and two were immunocompetent patients. The CMV reactivation occurs frequently in critically-ill immunocompetent patients and has been associated with immune paralysis, higher risk of other NIs, longer length of ICU stay, and death [29,30].

Other noninfectious major complications occurring during ECMO were also frequent in our patients, such as thrombocytopenia occurring in 50.7%, hypoxemia in 47.8%, hemorrhage in 29.9%, acute kidney failure in 29.4%, and limb ischemia in 23.9% of patients. None of them were related to nosocomial infections.

In this study, the overall mortality was 47.8%, and the mortality under ECMO was 39.1%. Nevertheless, we failed to demonstrate that nosocomial infections were associated with a greater mortality. Maybe the low rate of the MDR isolate, the application of the bundle of infection control, and the antimicrobial stewardship programs aiming to provide optimal antimicrobial therapy to patients quickly after an infection is suspected partly explain our results. The basal creatinine value and lactate levels within the first 4 h after ECMO initiation were predictive of mortality during ECMO. These results may be partly explained as a proxy for the severity of the underlying condition requiring the VA-ECMO support and the baseline comorbidities [31].

We estimate that the mortality attributable to VAP in our patients was 13%, consistent with other studies [32,33].

The present study might be limited by the fact that it was a single-center study, retrospective, and of a small sample size, making difficult the comparison of the weight of the contribution of each risk factor. In addition, the low sample size might have determined the lack of power, failing to achieve statistical significance. Furthermore, differences due to underlying heart failures requiring VA-ECMO, with different degree of severity, may have acted as confounders on the mortality.

## 5. Conclusions

Despite several limitations, our study provides relevant findings given the scarcity of studies that have explored nosocomial infections during VA-ECMO in CICUs. Our findings highlight the need for preventive measures aimed at reducing nosocomial infection rates and implementing stewardship strategies to optimize the use of antimicrobials.

Prospective and multicentric studies are warranted to implement the role of clinical and microbiological surveillance according to the timing of ECMO implant to reduce the infection-related morbidity and mortality.

## Figures and Tables

**Figure 1 microorganisms-11-01079-f001:**
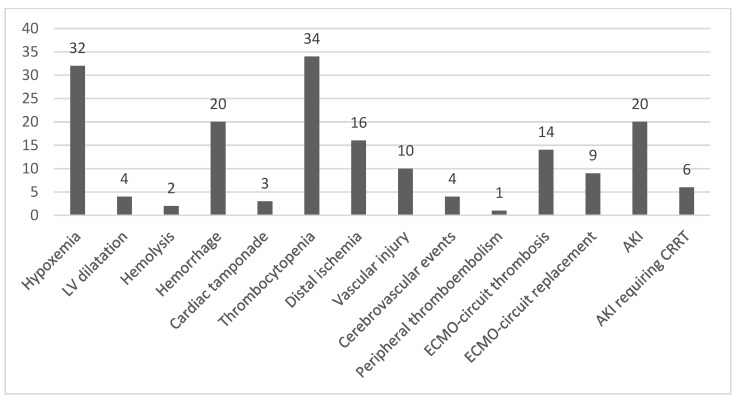
Noninfectious complications occurring during extracorporeal membrane oxygenation.

**Figure 2 microorganisms-11-01079-f002:**
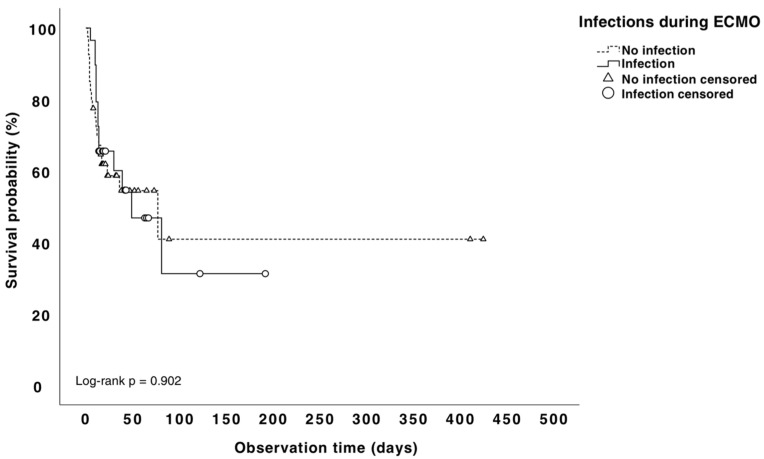
Kaplan–Meier mortality rates for patients with and without nosocomial infection.

**Table 1 microorganisms-11-01079-t001:** Basal characteristics and clinical and analytical data in patients with and without nosocomial infections.

Variable	Total (*n* = 69)	Noninfected = 40 (58%)	Infected = 29 (42.0%)	*p*
Baseline Characteristics				
Sex, male (%)	57 (82.7%)	31 (77.5)	26 (89.7)	0.218
Age, median years (IQR)	58 (50.5–62.8)	59 (50–64.5)	55 (50.6–61.1)	0.491
Underlying conditions (%):				
BMI > 25 kg/m^2^	35 (50.7)	20 (50)	15 (51.7)	0.999
Arterial hypertension	33 (49.3)	16 (42.1)	17 (58.6)	0.222
Cigarette smoking *	26 (38.8)	12 (31.6)	14 (48.3)	0.365
Stopped smoking > 5 years	13 (19.4)	9 (23.7)	4 (13.8)	0.365
Dyslipidemia	27 (40.3)	13 (34.2)	14 (48.3)	0.317
Peripheric vascular disease	7 (10.4)	4 (10.5)	3 (10.3)	0.999
Cardiac disease	52 (75.4)	25 (62.5)	27 (93.1)	0.004
Chronic pulmonary disease	8 (11.8)	5 (12.5)	3 (10.7)	0.999
Diabetes mellitus	19 (27.5)	9 (22.5)	10 (34.5)	0.290
Liver cirrhosis	1 (1.4)	0	1 (3.4)	0.420
Chronic kidney failure	9 (13)	3 (7.5)	6 (20.7)	0.152
Chronic renal replacement	0	05 (12.5)	0	-
Immunosuppression:	10 (14.5)		5 (17.2)	0.732
HIV infection (<200 CD4+)		0		
Solid tumor with active CT	0	0	0	-
HT neoplasia/HSCT (<5 yrs)	0	3 (7.5)	0	-
Solid organ transplant	3 (4.3)	0	0	0.258
IS therapy/corticosteroids	2 (2.9)	2 (5)	2 (6.9)	0.173
Autoimmune disease	6 (8.7)	2 (5)	4 (13.8)	0.230
	5 (7.2)	0	3 (10.3)	0.643
	0		0	-
Clinical and analytical data before ECMO				
Lactic acid prior to ECMO	8.1 (4.9–12.2)	8.7 (4.75–13.6)	7.6 (5–10.7)	0.202
Serum creatinine prior to ECMO	1 (0.9–1.3)	0.9 (0.8–1.2)	1.2 (0.9–1.6)	0.027
Days of MV before ECMO	1.3 (0.6–2.0)	1.3 (0.9–1.9)	2 (0.5–5.5)	0.930
Ejective fraction (%) prior to ECMO	18.1 (11.3–30.8)	19.1 (10.8–34.2)	17.3 (11.7–25.6)	0.374
RBC requirement before ECMO (≥5 U) (%)	3 (4.3)	3 (7.5)	0	0.258
Type of failure (%)				
Cardiac	62 (89.9)	36 (90)	26 (89.7)	0.999
Cardiac + respiratory	5 (7.2)	3 (7.5)	2 (6.9)	0.999
Respiratory	2 (2.9)	1 (2.5)	1 (2.5)	0.999
ECMO indication:				
Cardiogenic shock	38 (55.1)	21 (52.5)	17 (58.6)	0.634
Myocarditis	1 (1.4)	1 (2.5)	0	0.999
End-stage heart failure	0	0	0	-
Cardiac arrest	21 (30.4)	11 (27.5)	10 (34.5)	0.601
Pulmonary embolism	8 (11.6)	6 (15)	2 (6.9)	0.453
Arrhythmic storm	13 (18.8)	8 (20)	5 (17.2%)	0.999
ECMO as bridge to heart Tx (%)	9 (13%)	5 (12.5)	4 (13.8)	0.999
Cardiac arrest in the last 24 h (%)	44 (66.7)	26 (70.3)	18 (62.1)	0.600
SOFA score	7.9 (5.8–10.4)	8.2 (6.30–10.0)	7.0 (5.4–10.4)	0.537
Charlson comorbidity score	2.8 (1.9–3.7)	3 (2.0–4.0)	2.7 (1.86–3.6)	0.557
Major surgery before ECMO (%)	2 (2.9)	2 (5)	0	0.506
Colonization before ECMO (%)	21 (30.4)	10 (25)	11 (37.9)	0.295
MDR colonization	0	0	0	-
Any infection before ECMO (%)	10 (14.5)	9 (22.5)	1 (3.4)	0.037
Antibiotics >24 h before ECMO (%)	7 (5.2)	2 (5)	3 (10.3)	0.643

BMI: body mass index, IS: immunosuppressive, CT: chemotherapy, Yrs: years, HSCT: hematopoietic stem cell transplant, HT: hematologic neoplasia, MV: mechanical ventilation, RBC: red blood cells, Tx: transplant.

**Table 2 microorganisms-11-01079-t002:** Clinical and analytical data between patients with and without nosocomial infections supported by VA-ECMO during extracorporeal membrane oxygenation implantation.

Clinical and Analytical Data during ECMO Support	Total (*n* = 69)	Noninfected = 40 (57.9%)	Infected = 29 (42.0%)	*p*
BCIAO (%)	51 (76.1)	26 (68.4)	25 (86.2)	0.147
Peripheric cannulation (%)	59 (86.8)	35 (89.7)	24 (82.8)	0.481
Cannulation time (minutes) SD	25.7 (20.2–31.8)	25.7 (20.3–33.1)	25.71 (20.0–30.9)	0.245
Days on ECMO	6.30 (4.6–9.4)	5.73 (5.7–7.7)	7.20 (5.4–10.4)	0.053
Complications during ECMO	41 (61.2%)	19 (46.3%)	22 (53.7%)	0.044
Type of complications during ECMO (%)				
Cardiac failure	44 (63.8)	20 (50%)	24 (82%)	0.006
Hypoxemia	32 (47.8)	17 (44.7)	15 (51.7)	0.627
LV dilatation	4 (6.0)	1 (2.6)	3 (10.3	0.308
Hemolysis	2 (3.0)	1 (2.6)	1 (3.4)	0.999
Hemorrhage	20 (29.9)	8 (21.1)	12 (41.4)	0.106
Cardiac tamponade	3 (4.5)	2 (5.3)	1 (3.4)	0.999
Thrombocytopenia	34 (50.7)	21 (55.3)	13 (44.8)	0.464
Limb ischemia	16 (23.9)	9 (23.7)	7 (24.1)	0.966
Vascular injury	10 (14.9)	5 (13.2)	5 (17.2)	0.736
Cerebrovascular events	4 (6.0)	0	4 (13.8)	0.031
Peripheral thromboembolisms	1 (1.5)	0	1 (3.4)	0.433
Thrombosis of the circuit	14 (20.9)	5 (13.2)	9 (31)	0.128
ECMO circuit replacement	9 (10.8)	4 (10.8)	5 (17.9)	0.483
Acute kidney failure	20 (29.4)	12 (30.8)	8 (27.6)	0.796
In CRRT	6 (8.7)	3 (7.5)	3 (10.3)	0.690
Antiplatelet therapy (%)	48 (72.7)	27 (71.1)	21 (75)	0.785
ASA	3 (4.5)	3 (7.9)	0	0.256
ASA + Clopidogrel	3 (4.5)	0	3 (10.7)	0.072
ASA + Dipiridamol	40 (60.6)	24 (63.2)	16 (57.1)	0.799
Clopidogrel	2 (3.0)	0	2 (7.1)	0.176
Switch to C-ECMO (%)	5 (7.5)	1 (2.6)	4 (13.8)	0.158
Lactic Acid (2 h), mg/dL, median	5.8 (3.6–9.2)	6.0 (6.0–9.9)	5.3 (3.7–6.5)	0.249
Lactic Acid (4 h), mg/dL, median	2.03 (1.5–4.0)	2.13 (1.5–4.4)	1.8 (1.3–3.2)	0.670
Colonization of ECMO cannula(%)	2 (2.9)	0	2 (6.9)	0.173
Outcome				
Switch from ECMO to another CV support (%)	11 (15.9)	5 (12.5)	6 (20.7)	0.507
In-hospital stay, days	20.3 (13.1–45.0)	21.7 (11.3–45.0)	19.3 (14.1–45.3)	0.980
ICU stay, days	14 (9.5–25)	12.4 (7.6–23.3)	15.7 (10.9–34.3)	0.860
Mortality during ECMO (%)	27 (39.1)	18 (45)	9 (31)	0.319
In-hospital mortality (%)	33 (46.4)	18 (46.2%)	15 (50.0%)	0.754

ASA: aminosalicylic acid, BCIAO: intra-aortic balloon, C-ECMO: central ECMO, CRRT: continuous renal replacement therapy, ICU: intensive care unit, LV: left ventricular, MDR: multidrug resistant, SD: standard deviation.

**Table 3 microorganisms-11-01079-t003:** Site of infection and associated microorganisms.

Number of NI Episodes during ECMO	N = 34 (%)
Ventilator-associated pneumonia-associated organisms	19 (55.9%)
Gram-negative bacteria	8
*Klebsiella pneumoniae*	2
*Proteus mirabilis*	1
*Serratia marcescens*	1
*Burkholderia cepacia*	1
*Enterobacter cloacae*	1
*Enterobacter aerogenes*	1
*Acinetobacter baumannii*	1
Gram-positive bacteria	5
*S. aureus*	3
MRSA ^a^	1
Polymicrobial ^b^	3
Virus	1
HSV-1 ^c^	1
Unknown etiology (negative culture)	2
Tracheobronchitis	3 (8.8%)
*Enterobacter aerogenes*	2
*Stenotrophomonas maltophilia*	1
Bloodstream infections	3 (8.8%)
*Pseudomonas aeruginosa*	1
Coagulase-negative staphylococci	2
SSTI ^d^	3 (8.8%)
*Morganella morganii*	1
*Staphylococcus lugdunensis*	1
Coagulase-negative staphylococci	1
Intra-abdominal infections (*Escherichia coli*)	1 (2.9%)
*Clostridium difficile infection*	1 (2.9%)
Urinary tract infection (*K. pneumoniae*)	1 (2.9%)
CMV ^e^ disease	3 (8.8%)

^a^ Methicillin-resistant *Staphylococcus aureus*. ^b^ >1 pathogen known to be responsible for nosocomial pneumonia; ^c^ Human herpes virus 1. ^d^ is the abbreviation of skin and soft tissue (SSTI) infections, ^e^ is the abbreviation of cytomegalovirus (CMV).

**Table 4 microorganisms-11-01079-t004:** Univariate analysis of the risk factors for mortality.

Variable	N (%)	Alive = 36	Death = 33	*p*
Sex, male	57 (82.6%)	31 (86.1%)	26 (78.8%)	0.423
Age (median, IQR)	58 (IQR 20–75)	55 (20–75)	61 (27–69)	0.416
Body mass index > 25	33 (47.8)	14 (38.9%)	19 (57.6%)	0.12
Underlying condition				
Cardiac disease	52 (75.4%)	25 (69.4%)	27 (81.8%)	0.23
Pulmonary disease	8 (11.6%)	4 (11.1%)	4 (12.1%)	1.00
Diabetes mellitus	18 (26.1%)	11 (30.6%)	7 (21.2%)	0.37
Liver cirrhosis	1 (1.4%)	1 (2.8%)	0 (0%)	1.00
Chronic kidney insufficiency	8 (11.6%)	3 (8.3%)	5 (15.2%)	0.466
Chronic renal replacement	0 (0%)	0 (0%)	0 (0%)	–
Immunosuppression	16 (10.1%)	3 (8.3%)	4 (12.1%)	0.70
HIV infection	0	0 (0%)	0 (0%)	–
Solid organ tumor under CT	0	0 (0%)	0 (0%)	–
Hematol. neoplasia/HSCT (<5 yrs)	3 (4.3%)	2 (5.6%)	1 (3.0%)	1.00
Solid organ transplant	2 (2.9%)	1 (2.8%)	1 (3.0%)	1.00
Immunosuppressive therapy/CCS *	6 (8.7%)	2 (5.6%)	4 (12.1%)	0.41
Autoimmune disease	5 (7.2%)	3 (8.3%)	2 (6.1%)	1.00
ECMO indication				
Cardiocirculatory failure	60 (87.0%)	35 (97.2%)	25 (75.8%)	0.008
As bridge to heart Tx	9 (13%)	9 (25%)	0 (0%)	0.002
Respiratory failure	4 (5.8%)	0 (0%)	4 (12.1%)	0.047
Cardiorespiratory failure	5 (7.2%)	1 (2.8%)	4 (12.1%)	0.18
Type of cannulation				
Veno-venous	2 (2.9%)	0	2 (6.7%)	0.18
Venous artery	66 (95.7%)	38 (97.4%)	28 (93.3%)	0.57
Other	1 (1.4%)	1 (2.6%)	0	1.00
Femoral cannulation	66 (95.7%)	39 (100%)	27 (90%)	0.70
Days on ECMO	6 (5–35)	6 (2–26)	7 (2–35)	0.14
Renal replacement therapy	6 (8.7%)	1(2.8%)	5(15.2%)	0.097
Days of MV before ECMO	1 (1–20)	1 (1–6)	1 (1–20)	0.28
Total days of MV	1 (1–2.7)	1 (1–6)	1 (1–20)	0.28
SOFA score	8 (2–17)	7.5 (2–13)	9 (13–17)	0.051
Charlson score	3 (0–7)	2 (0–6)	3 (0–7)	0.14
Colonization before ECMO	21 (30.4%)	10 (27.8%)	11 (33.3%)	0.61
MDR colonization	1 (1.4%)	0 (0%)	1 (3.0%)	0.47
Infections due to the same microorganisms during ECMO	0	0 (0%)	0 (0%)	
ECMO cannula colonization	2 (2.9%)	2 (5.6%)	0 (0%)	0.49
Microorganism				
*Propionibacterium*	1 (1.4%)			
*Enterobacter cloacae*	1 (1.4%)			
Other VAD requirement	4 (5.8%)			
Antibiotics > 24 h before ECMO	6 (8.7%)	2 (5.6%)	4 (12.1%)	0.41
Adequate empirical antimicrobial therapy	12 (17.4%)	6 (16.7%)	6 (18.2%)	0.86
Endpoints				
Length of in-hospital stay	20 (4–411)	36 (9–411)	14 (4–78)	0.01
Length of CICU stay	14 (2–405)	15 (5–405)	12 (2–49)	0.15
Mayor surgery before ECMO	4 (5.8%)	1 (2.8%)	3 (9.1%)	0.34
BSI or VAP related mortality	5 (7.2%)	–	–	–
Mortality after ECMO weaning	28 (40.6%)	–	–	–
In-hospital mortality	33 (47.8%)	–	–	–

CT: chemotherapy; HSCT: hematopoietic stem cell transplant; Tx: transplant; CCS: chronic corticosteroid therapy; MV: mechanical ventilation; CICU: cardiac intensive care unit; VAD: ventricular assist device.

## Data Availability

No new data were created or analyzed in this study. Data sharing is not applicable to this article.

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
