# Peer review of "Nosocomial Infections in Adult Patients Supported by Extracorporeal Membrane Oxygenation in a Cardiac Intensive Care Unit"

_microorganisms, 2023, doi:10.3390/microorganisms11041079_

Round 1

Reviewer 1 Report

I have reviewed the manuscript entitled: Nosocomial Infections in Adult Patients Supported by Extra corporeal Membrane Oxygenation in a Cardiac Intensive Care Unit. This is very interesting manuscript describing rare patients group. You write: "Continuous variables were reported as mean SD." (line 117) however the continous data you report in anotherway (line 133 : Median age was 58 years (IQR 50.5 – 62.8)) - please correct it. Please review Table 1 - it is something wrong with lines with data, it is very unreadable. Please describe exactly your antibiotic prophylaxis protocol for ECMO cannulation i.e. doses, repeat time, duration of prophylaxis. Was any extra doses applied in case of excessive bleeding etc.? Please discuss also antibiotic empirical therapy protocol you have used and what, if any, changes in such a protocol you would recommend based on your results? Thank you for this interesting manuscript with good data report, which in my opinion is worth publishing.

Author Response

Thank you, we appreciate the time you and the other reviewers have dedicated to providing valuable feedback on my manuscript. I'll provide a track mode version of the revised version of the Manuscript, so please see the attachment. Hereafter, I reply(in red) to your comments (in black).

Point 1: You write: "Continuous variables were reported as mean SD." (line 117) however the continous data you report in another way (line 133 : Median age was 58 years (IQR 50.5 – 62.8)) - please correct it. Response: We thank the reviewer for pointing this out.

Response: We have revised the text by modifying it from "mean and SD" to the median and interquartile range.

Point 2: Please review Table 1 - it is something wrong with lines with data, it is very unreadable.

Response: I agree with this and have reformatted Table 1.

Point 3: Please describe exactly your antibiotic prophylaxis protocol for ECMO cannulation i.e. doses, repeat time, and duration of prophylaxis. Were any extra doses applied in case of excessive bleeding etc.? Please discuss also antibiotic empirical therapy protocol you have used and what, if any, changes in such a protocol you would recommend based on your results.

Response: We have, accordingly to your suggestion (lines 75-80), further explained the antimicrobial stewardship procedures that we didn't have previously detailed due to the word limit. We don’t extend the antimicrobial prophylaxis in case of bleeding without systemic signs of infection. We have an internal antimicrobial guide for clinical use. This manual provides a guide to choosing an empiric antibiotic treatment in different situations, such as b-lactam allergy or previous MRSa colonization etc.... About this protocol, You have raised an important point here. However, this was an epidemiological study in a very particular setting, the cardiac intensive care unit, with the intent of exploring nosocomial infection in people with severe cardiac failure we found that pneumonia is very frequent, but our study was not aimed to explore the treatment effectiveness. In my opinion, our study suggests that we would raise attention towards ventilation-related pneumonia, and in case of suspicion, antimicrobial treatment should cover gram-negative bacteria.

Reviewer 2 Report

Thank you for the opportunity to review this interesting and important manuscript

I have following questions and suggestions:

Comments to abstract:

1.       “From 69 patients treated with VA-ECMO >48 hours, (median age 58 years). 29 (42.0%) 26 patients developed 34 episodes of infections with an infection rate of 0.92/1,000 ECMO days.” In my opinion the description of infection rate is complicated, should be explained how it was calculated or be removed. Comma should be removed in 1000.

2.        “In-hospital mortality was 47.8%, but no association with nosocomial infections was found 30 (P= .75).” I was intrigued that infections did not increase mortality in the ECMO-population. Is the calculation correct?

Comments to methods:

3.       Description of the CICU would be important. How many patients per room? What is the distance between beds? How many patients are treated by the same nurse? What kind of ventilation? How is the catheterization for ECMO done, in the CICU? In the OR? In a cath lab? Are the infected ECMO-patients isolated? It would be of great interest if those issues were discussed in the discussion part, how to decrease the hazard for infections in the ECMO-population.

4.       Indications for ECMO is not mentioned.

5.       What does it mean: This study was in accordance with the ethical standards of the local institution’s Committee (MICRO.HGUGM.2019-010) approved on July 4th. 2019. Was the study approved by the ethical committee? Please clarify.

6.       How were the variables from the univariate analysis selected for the multivariate? What was the structure of the multivariate analysis?

Comments to discussion:

7.       See above under methods: it would be of great interest if the impact of the environment in the CICU is discussed, ventilation, number of patients in the same room, how is the catheterization for ECMO done, in CICU, in the OR, in a cath lab? How are the patients with infections treated? How can the number of infections be decreased in this very sensitive group of patients?

Author Response

Thank you for giving me the opportunity to submit a revised draft of my manuscript. I'll provide a track-mode version of the Manuscript with the revision, so please see the attachment. I'll reply to your questions (in black) with our responses (in red).

Point 1:  “From 69 patients treated with VA-ECMO >48 hours, (median age 58 years). 29 (42.0%) 26 patients developed 34 episodes of infections with an infection rate of 0.92/1,000 ECMO days.” In my opinion the description of infection rate is complicated, should be explained how it was calculated or be removed. Comma should be removed in 1000.

Response: Thank you for this suggestion, we explained how we calculated the infection rate on lines 116-118. However, we believe this definition is appropriate and should be retained because it is the most commonly applied among ECMO studies as well the comma. Hereafter the most valuable papers that use the same definition. (Int J Antimicrob Agents. 2017 Jul;50(1):9-16. doi: 10.1016/j.ijantimicag.2017.02.025. Epub 2017 May 18. PMID: 28528989.      Pediatr Crit Care Med. 2011 May;12(3):277-81. doi: 10.1097/PCC.0b013e3181e28894. PMID: 20495508.   Ann Transl Med. 2018 Nov;6(21):427. doi: 10.21037/atm.2018.10.18. PMID: 30581835; PMCID: PMC6275411.)

Point 2  “In-hospital mortality was 47.8%, but no association with nosocomial infections was found 30 (P= .75).” I was intrigued that infections did not increase mortality in the ECMO-population. Is the calculation correct?

Response: We have, accordingly, revised the data: the result has been confirmed. Due to the small sample size and the severity of the underlying disease, there are many confounding factors. Of note, the majority of patients were rescued under an urgency regime and placed in ECMO within 72h of the admittance because of the severity of illness: this particular setting (cardiac intensive care unit)population is quite different from the classical ECMO population and baseline risk factor for mortality is extremely high. 

Point 3 Description of the CICU would be important. How many patients per room? What is the distance between beds? How many patients are treated by the same nurse? What kind of ventilation? How is the catheterization for ECMO done, in the CICU? In the OR? In a cath lab? Are the infected ECMO-patients isolated? It would be of great interest if those issues were discussed in the discussion part, how to decrease the hazard for infections in the ECMO population.

Response: thanks for pointing this out. We have revised the Manuscript following your suggestion on lines 65-66 and 70-72. In brief, in CICU there is only one patient per room, and patients are all subjected to invasive mechanical ventilation; nevertheless, the precise setting of the ventilator was not addressed because it is beyond the scope of this paper. On lines 287-289, following your suggest, we have included a possible explanation of our results.

Point 4     Indications for ECMO is not mentioned. Response: On lines 59-60, we report as follows "All adults (>18 years) receiving VA-ECMO >48h for cardiac arrest or severe cardiogenic shock from January 2013 to December 2018 within the cardiovascular intensive care unit (CICU) were evaluated for inclusion in the study". Therefore indication for ECMO was cardiogenic shock and cardiac arrest.

Point 5 What does it mean: This study was in accordance with the ethical standards of the local institution’s Committee (MICRO.HGUGM.2019-010) approved on July 4th. 2019. Was the study approved by the ethical committee? Please clarify. Response: Thank you for pointing this out. Our study was approved by Ethics Committee for Healthcare of Gregorio Marañón Hospital (https://www.iisgm.com/documentacion-ceim/) to advise on possible conflicts of an ethical nature that could derive from clinical care practice. This abbreviation (MICRO.HGUGM.2019-010) refers to the file number by which our study protocol was approved. 

Point 6 How were the variables from the univariate analysis selected for the multivariate? What was the structure of the multivariate analysis? Response: Thanks for this question; on multivariate analysis for nosocomial infection and mortality we selected variables that resulted significative or nearly significative(P<0.05) at univariate analysis. We used a regression model (linear and logistic). We didn't report a table because we should have added another table to the Manuscript

Point 7 See above under methods: it would be of great interest if the impact of the environment in the CICU is discussed, ventilation, number of patients in the same room, how is the catheterization for ECMO done, in CICU, in the OR, in a cath lab? How are the patients with infections treated? How can the number of infections be decreased in this very sensitive group of patients? Response: We agree with this comment, we have extended the methods (lines 62-66 and 71-78) by explaining our infection control bundle and the organization of CICU, which we previously omitted for exceeding the word limit. On lines 292-295 we added the relevance of infection control and bundles to limit the incidence of nosocomial infections.